**Cluster analysis of multimorbidity and healthcare burden based on machine learning: results from CHARLS**

**Background:**

With the ageing of the worldwide population, the phenomenon of chronic noncommunicable diseases, particularly the increasing prevalence of multimorbidity. Around one-third of the global adults is afflicted with multiple diseases, with approximately a quarter of in China exhibit multimorbidity. Individuals suffering multiple medical conditions face a higher risk of early death, extended hospital stays and higher expenses compared to those with a single condition. It's common knowledge that multimorbidity varies with age, leading to a greater occurrence among the middle-aged and elderly, necessitating increased societal focus. Such patterns underscore the importance of sufficient health care availability during middle and later stage of life. Therefore, multimorbidity of the health care profession is a huge challenge worldwide, placing a particularly heavy burden on both individuals and society.

The underlying mechanisms of multimorbidity development are intricate, interconnected, and involve the interdisciplinary interaction of multiple disciplines[4]. This implies that there is considerable heterogeneity in multimorbidity, which means that multimorbidity based on a combination of different diseases will have distinct impacts and medical burdens for individuals. Recognizing and profiling patients with multiple diseases sharing similar patterns can alleviate healthcare load, a method adding in the creation of efficient early diagnosis and prevention of numerous diseases, along specific interventions. However, a comprehensive system for managing multiple concurrent conditions and guidelines is absent. Consequently, segregating individuals with akin illnesses from those with multiple complex conditions is essential for more targeted treatment and care.

Previously, the majority of methods used for cluster analysis were exploratory approaches, including factor analysis and hierarchical clustering. Currently, a one-size-fits-all approach for charactering multiple diseases does not exist, and diverse techniques exist for analyzing clusters of multimorbidity. The application of clustering approaches to multimorbidity may also be valuable. For instance, multiple correspondence analysis is employed for grouping and examining urban groups in London, while t-distributed Stochastic Neighbor Embedding (tSNE) and uniform manifold approximation and projection (UMAP) methods are infrequently utilized for such clustering.

We aimed to pinpoint and validate clusters of individuals with multimorbidity on different Chinese regions, as well as the connection to healthcare burden and mortality, and provide tailored clinical strategies and preventative actions to support clinicians and health care.

**Methods**

**Study Population**

The data used in this study are from the Harmonized China Health and Retirement Longitudinal Study (CHARLS), a prospective and national population-based cohort study. The CHARLS survey project was approved by the Ethics Committee of Peking University and all participants have been anonymized. A total of 25586 individuals participated in 2011, and three waves of follow-up had been conducted in 2013, 2015, and 2018. The detailed description of the CHARLS study design has been reported previously. This study followed the Strengthening the Reporting of Observational Studies in Epidemiology (STROBE) reporting guideline for cohort studies.

The objective in this study was to track the multimorbidity and adverse outcome during the follow-up, we chose the population data in 2011 as the baseline and the following three waves visits as follow-up. We enrolled the participants who had multimorbidity in 2011 as the target population, therefore, 3 475 participants with no multimorbidity in 2011 (< 2 chronic diseases) have been excluded. The characteristics of participants with and without multimorbidity at baseline are provided in eTable1 in the Supplement. We also excluded the individuals who had blank information in 2011 (n=7 878). In addition, 441 participants died in 2011 were excluded. Ultimately, there were 13 792 individuals were included in this study (Figure e1). Participants provided sociodemographic characteristics (age, sex, education level, household income, and health insurance patterns), lifestyle (smoking, drinking, and living habits), health-related history (hospital admissions) information and underwent physical examinations and laboratory examinations at baseline.

**Definition of multimorbidity**

Multimorbidity was defined as the simultaneous presence of two or more diseases or health conditions in the same individual.[2] In this study, 19 diseases and health conditions were collected at baseline: hypertension, diabetes, obesity, hyperuricemia, dyslipidemia, cancer, lung disease, heart problem, stroke, psychiatric problem, arthritis, liver disease, kidney disease, stomach/digestive disease, asthma, memory problem, depressive symptoms, cognitive impairment, and amaemia. We ascertained these diseases and health conditions based on self-reports of a physician's diagnosis, the assessment of questionnaire items, medication data, and the results of laboratory test (eMethod in

Supplement).

**Outcomes**

The outcomes of interest were all-cause mortality and health care burden (four items about hospital nights last recent time, hospital stay times a year, hospitalization out-of-pocket expenditure and hospitalization total expenditure a rear). Death/ mortality were ascertained through the record of death information in the Harmonized CHARLS datasets.

**Statistical analysis**

Statistical analysis was performed from September 2023 to May 2024. Categorical variables were expressed in terms of frequencies and precentages, variables of continuous normal distribution were showed as mean with standard deviation and continuous skewed distribution as medians with interquartile intervals (IQI). For comparisons of characteristics across different groups, analysis of variance was used for continuous variables and χ2 tests were applied for categorical variables. Given the improbability of complete documentation of the disease history, we approach the few absent disease state as though the patient had no disease. Additionally, certain variables like BMI, blood glucose, blood lipids, and hemoglobin are absent, and we regard these missing figures as within the standard range. Statistical analyses were performed using R version 4.3.2, Python 3 ,GrapPad Prism 9.5.1 and OriginPro 2021. A 2-sided $p$ value of less than 0.05 was represented statistically significant.

T-distributed Stochastic Neighbor Embedding (tSNE) was performed to reduce the dimensionality of data through machine learning and focus on the most important features. Then k-means were used to identify clusters of multimorbidities (eFigure 2 in Supplement). K-means algorithm is a clustering technique that exhibits a stronger ability to separate different clusters. We ran the algorithm on 3 to 8 possible clusters as we deemed a lager number of clusters are not clinically significant. The most appropriate number of clusters was considered using the composite indices of likelihood ratio test (LRT), Davies Bouldin, calinski Harabase and Silhouette Coefficent (eFigure 3 in Supplement).

After the optimal clusters was selected, characteristics were described accroding to each multimorbidity cluster, and the frequencies and percentages of disease and health conditions in each cluster were claculated. The observation/expected (O/E) ratio was calculated as the prevalance of disease in each cluster group divided by the total prevalance. The nomenclature of prominent condictions for each cluster was identified by the intra-cluster prevalence ≥ 20% and O/E ratio ≥ 2.

We calculated crude rates of incident death events per cluster and expressed them per 1000 person-years at risk. Survival probabilities in the clusters were calculated using the Kaplan-Meier method, were compared using the log-rank test, and the Cox proportional hazard models were used to estimate hazard ratios (HRs) and 95% CIs for the associations between distinct cluster groups and the risks of death, with unspecific populations as the reference group. Model 1 was adjusted for age and sex. Model 2 was furthur adjusted health risk factor, including smoking, drinking, medication use, BMI category, medication use, and place of residence (rural or urban).

To evaluate the link between hospital admissions and various clusters and stratified groups, negative binomial regression models were employed, addressing the disproportionate spread of hospitalizations. Odds ratios were calculated using logistic regression to examine the cluster differences in mortality rates at the end of the second, fourth and seventh years.

To confirm the robustness of the clustering effect, we initially removed the disease with the delection, then performed multiple imputation on absent data. Ultimately, to check the consistency of the results, the uniform manifold approximation and projection (UMAP) approach was employed to simplify the data's dimensionality, followed by the use of k-means to segment the data into various cluster groups. We also performed stratified analyse for different age, gender and medicine use.

**Results**

**Baseline characteristics**

We identified 13,792 individuals with multimorbidity between January 1, 2011 and November 11, 2018. The median age was 58 years (inter-quartile range 51–65) and 46.61% were male. Cognition impairment (60.65%) was the most common comorbidity, followed by hypertension (41.79%), arthritis (34.03%), depression (32.55%) and dyslipidemia (30.08%) and digestive disease (23.74%) make up the top six most prevalent conditions (Figure 1). Relative to individuals without multimorbidity, those with multimorbidity were older, exhibited elevated BMI, both systolic and diastolic blood pressure, possessed less educational attainment, lived alone more, were more rural, had higher glucose, TC, triglyceride, LDL and UA, and had lower HDL(Table S1).

**Figure 1** Prevalence of long-term conditions in the multimorbidity and without multimorbidity conditions

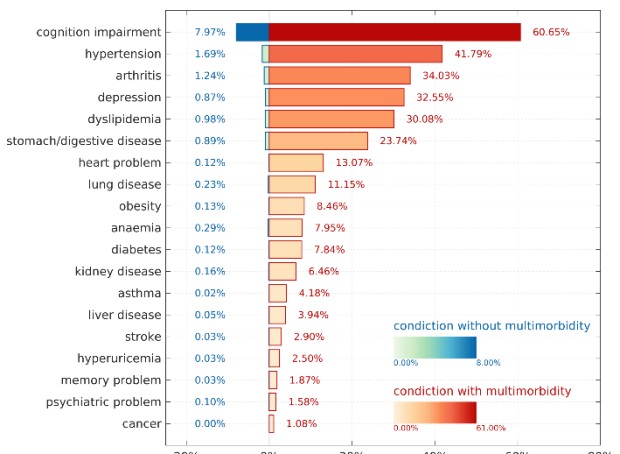

**Clustering of multimorbid individuals**

Following the assessment of k-means clustering using the fitting evaluation index, a four-group solution was pinpointed to describe comorbidity patterns (eFigure 1 in Supplement). The four clusters are characterized by diverse disease conditions and proportions, were named according to their main features: cancer, respiratory and digestion, hypertension and heart and digestion. The proportion of individuals with given conditions belongs to clusters in the whole sample was shown in Figure S4. The attributes of the four groupings were depicted in eTable 3, which were further described after data completion (eTable 2). In the respiratory and digestion group, men had the highest probability, whereas in the hypertension group, the variances in systolic, diastolic, pulse pressure and glucose were most pronounced. Heart and digestion group had a higher propensity to experience dyslipidemia. Figure 2 illustrated that the disease distribution was even in the low-burden, in contract to the more varied percentages in the respiratory and digestion group and the hypertension group.

**Figure 2 Each disease accounted for the precentage of the disease in every cluster group.** The numbers in the graph represented the incident in %.

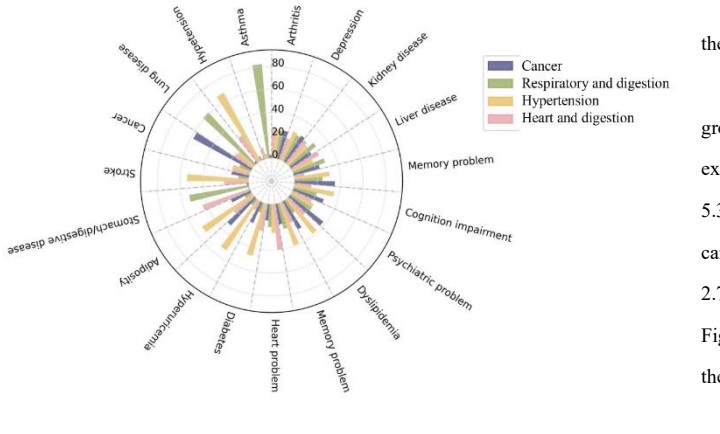

Using a Sankey diagram, approaches showed consistent (Figure 3). Within these cluster with heart and disgestive group most comorbidities, averaging per individuals, in contrast to the which had the fewest, averageing per person. The ROC graph in further indicated minimal risk in with cancer and maximal risk in hypertension group.

the two results groups, those exhibited the 5.32 diseases cancer group, 2.73 diseases Figure S5 the group the

**Figure 3 Number of diseases and classification of individuals by primary analysis method alternative approach.** The on the left indicate the number of multimorbidity, and the numbers on and right represent the cluster

**Correlations among various and Risk of death**

The correlation coefficient revealed an inverse relationship heart problem and both

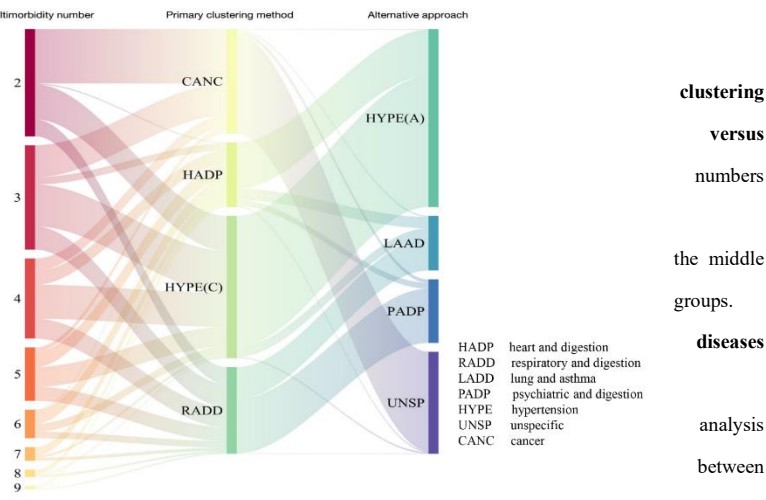

clustering versus numbers the middle groups. diseases analysis between

hypertension and diabetes, while dyslipidemia showed a positive correlation with diabetes, and asthma was linked to lung disease (Figure S6). When both diseases appeared concurrently, hypertension and cognitive impairment peaked at 30.98%, succeeded by cognitive impairment and arthritis at 24.12% (eTable 4).In Table 1, the correlation between the cluster groups and mortality risk was depicted, using the cancer group as a benchmark, with HR observed in groups with respiratory and digestion, hypertension, and heart and digestion group were 1.260 (1.137-1.397), 1.358 (1.240-1.489) and 1.258 (1.125-1.407), respectively. Subsequently, after adjusting for confounders, the risk was slightly decreased.

During the 7-year follow-up, 300 deaths (31.064 per 1000 person-years) occurred in cancer cohort and 377 deaths (39.790 per 1000 person-years) occurred in respiratory and digestion cohort. And the hypertension group had the highest RD mortality rate of 1.381 per 1000 person-years (95% CI, 1.260, 1.514).

**Table 1** Association of different group with mortality risk in 7-year follow-ups

| Hazard ratio (95% CI) | cancer | respiratory and digestion | hypertension | heart and digestion |
|---|---|---|---|---|
| **Unadjusted HR (95% CI)** | Ref. | 1.260 (1.137, 1.397) ** | 1.358 (1.240, 1.489) ** | 1.258 (1.125, 1.407) ** |
| **Model 1[a]** | Ref. | 1.183 (1.066, 1.313) * | 1.124 (1.024, 1.235) * | 1.075 (0.959, 1.205) |
| **Model 2[b]** | Ref. | 1.186 (1.040, 1.351) * | 1.235 (1.102, 1.384) ** | 1.151 (1.000, 1.325) |
| **Unadjusted HR[c] (95% CI)** | Ref. | 1.512 (1.194, 1.916) ** | 1.886 (1.536, 2.315) ** | 1.550 (1.203, 1.995) ** |
| **Model 1[ac]** | Ref. | 1.325 (1.043, 1.682) * | 1.387 (1.125, 1.710) ** | 1.215 (0.939, 1.751) |
| **Model 2[bc]** | Ref. | 1.189 (0.927, 1.525) | 1.331 (1.075, 1.648) ** | 1.118 (0.856, 1,461) |
| **Model 1[ad]** | Ref. | 1.183 (1.067, 1.312) ** | 1.133 (1.032, 1.243) ** | 1.082 (0.967, 1.212) |
| **Model 2[bd]** | Ref. | 1.138 (1.022, 1.266) * | 1.103 (1.003, 1.213) * | 1.027 (0.912, 1.156) |
| **Death, n (%)** | 300 (8.28) | 377 (12.55) | 763 (15.48) | 303 (13.56) |
| **Rate of death, per 1000 person-years** | 31.064 | 39.790 | 42.905 | 39.113 |
| **RD (95% CI), per 1000 person-years** | | 1.281 (1.156, 1.420) | 1.381 (1.260, 1.514) | 1.259 (1.126, 1.408) |

* p < 0.05 ** p < 0.01. [a] Adjusted for age, sex. [b] Adjusted for age, sex, smoking ever, drinking ever, BMI, medication use and live in rural or urban. RD, risk difference; [c] Sensitivity analysis 1: Delete the missing diseases. [d] Sensitivity analysis 2: Multiple imputation of data.

**Healthcare resource use**

Relative to the group with the low-burden, there was a notable enhancement in the use of medical resources across all multimorbidity clusters. In heart and digestion group, the frequency of one-year hospital stays, the latest hospitalization days, personal hospital stays, and one-year hospital stays topped the list, followed closely by the hypertension group in term of out-of-pocket expenses and total one-year hospitalization costs (Figure S7). Annually, the healthcare expenses for the hypertension cohort rose, exceeding those for those with respiratory and digestion group. With the leading 20 prevalent pairings, hypertension and heart disease stood out as the most burdensome, incurring an average yearly cost of 2030 yuan and 2.27 days of final hospital stay, with a 7-year death rate of 18.07%. A detailed stratification analysis was shown in Table S5, Table S6, Table S7.

**Discussion**

Our study identified four distinct multimorbidity clusters, namely the cancer, respiratory and digestion, hypertension, and heart and digestion, with varying patterns of healthcare utilization and mortality using routine sociological and clinical data from multiple regions. Importantly, these multimorbidity clusters had differential risks of healthcare burdens and death, which resulted in similar groupings in the validation clustering algorithm, suggesting that the multimorbid patterns reflect the anticipated outcomes of clinical symptoms across various disease pairings.

Previous studies found significant differences in the prevalence of multiple diseases. A systemic study found the prevalence of multimorbidity among individuals over 60 years ranged from 6.4% to 76.5%. Multimorbidity is associated with higher mortality rates, for example, some studies found a dose-response relationship between the number of diseases and mortality rates, with the risk of death more than three times higher than four multimorbidity. Similar to the research findings, our research revealed that the group with the highest average disease count for heart and digestion group had a higher mortality rate. Fan et.al found that participants with cardiometabolic multimorbidity had the highest risk of mortality (HR=2.20) after dividing the multimorbidity into four group patterns. Consistent with

previous studies, follow an average period of 7 years, we found the death rate peaked (HR=1.46) in the lung and asthma group, being 2.6 times greater than unspecific group (Table S8).

Addressing several illnesses is more intricate than managing just one, thereby amplifying the demand for medical resources. Treating these conditions requires tailoring for each patient, and comprehending the combination of expensive and affordable diseases allows us to more accurately identify multimorbidity patterns associated with high consumption, contributing to the creation of innovation intervention strategies. In our study, the primary cluster analysis revealed that groups with heart and digestion group consumed the most medical resources, with mortality trailing behind. In another algorithm, the groups with lung and asthma, despite utilizing the most medical resources, exhibited the greatest mortality rates. The results are similar to those in the city of London, where burden of cardiovascular disease is high.

Our research identified gender, age, BMI and smoking as the primary risk elements for multimorbidity mortality, in contract to alcohol intake, which showed no apparent correlation. Initially, female faced a higher likelihood of succumbing to multiple morbidities compared to male, but over time, the situation shifts. Within the group with hypertension, the mortality risk was greater for female compared to male, whereas in the respiratory and digestion and heart and digestion, male faced a heightened mortality risk than female. Studies have shown that a poor lifestyle can increase the risk of multimorbidity.

Notably, the cumulative burden and mortality has increased significantly for each cluster, but the rate of increase has not been uniform. Despite the hypertension group originally not facing the highest medical expenses, after seven years of development, out-of-pocket and total costs increased the fastest in this group compared to others, to 3.65 and 4.89 times the initial amount, respectively. This indicated that societal stress intensifies in the advanced phases of hypertension compared to the initial stages, underscoring the need to bolster hypertension prevention and management to lessen social and personal strain. This demonstrated a propensity among these individuals to accumulate multimorbidity, implying a higher probability of a third disease in those with two conditions, aligning with earlier research. A prospective study showed that individuals with both hypertension and diabetes had a higher risk of coronary heart disease or stroke than those with only one. This may be ascribed to the gradual accumulation of health side effects resulting from alternations in certain diseases over time. Pertaining to this, tailored health care and therapy for early multimorbidity, thereby enhancing the overall quality of life.

This study has several strengths. First, the advantage of this study is that a novel method is used to cluster multimorbidity individuals in multiple regions in China, presenting a new chronic disease model and adding new content to multimorbidity the development. In addition, the integration of clustering and machine learning-based analysis may establish a groundwork for the field of precision medicine. Despite the absence of a definitive gold standard in multimorbidity clustering, our work contributes to enhance robust and reproducible methods. Subsequently, we monitored the classified groups at different time points to ascertain the death rates and medical burden at each interval, along with diverse develop trends to comprehend the multimorbidity results. The finding of our study has significant implications for the management of multiple health conditions, the formulation of health policies, and the distribution of hospitals.

Nonetheless, our study results should be interpreted with the following limitations. First, no external validation was performed in this study. Second, there is no standardized way to measure multimorbidity. Third, the intrinsic constrains of the prevented us from conducting validation on a broader population. Fourth, the clustering of multimorbidity changes dynamically over time, but we solely consider clusters at certain time, neglecting the dynamic progression of diseases, potentially resulting in discrepancies between the follow-up and the actual disease conditions in time. Fifth, the focus of our research was predominantly on middle-aged and older individuals, with the younger demographic not grouped together. Fourth, despite employing imputation and removing absent dada, the lace of complete data may still influence the outcomes.

### Conclusions

Utilizing publicly available routine demographic and clinical data, our study identified four multimorbidity clusters with distinct clinical characteristics, healthcare use and mortality rates, primarily targeting middle-aged and elderly. The pathophysiological mechanisms that lead to these results, as well as the implications of potential interactions between individual diseases, still need to be confirmed in future studies. Our findings provide important evidence to guide clinical intervention and health care, as well as to optimize outcomes for patients with multimorbidity.

# Supplementary content

## eMethods. Supplemental Methods

### Definition of Diseases and Health Conditions

In this study, 19 diseases and health conditions were collected.

Hypertension was defined as self-reported hypertension history, and/or on anti-hypertension drugs, and/or mean systolic blood pressure (SBP) $\geq$140 mmHg, and/or mean diastolic blood pressure (DBP) $\geq$ 90 mmHg in all three measurements. Diabetes was defined in accordance with the American Diabetes Association criteria: random plasma glucose $\geq$ 11.1 mmol/L; HbA1c$\geq$6.5.

Obesity was identified as body mass index (BMI; calculated as weight in kilograms divided by height in meters squared) >28 for Chinese adults. Hyperuricemia was defined as the serum uric acid (UA) level >7.0 mg/dL. Dyslipidemia: total cholesterol (TC) $\geq$ 6.2 mmol/L, and/or triglyceride (TG) $\geq$ 2.3 mmol/L, and/or high-density lipoprotein cholesterol (HDL-C) $\leq$ 1.0 mmol/L, and/or low-density lipoprotein cholesterol (LDL-C) $\geq$ 4.1 mmol/L, and/or currently using lipid-lowering agents, and/or previously diagnosed with dyslipidemia by a doctor. Anaemia was defined by sex-specific cut-off values (haemoglobin < 130 g/L for male, < 120 g/L for non-pregnant female).

Depression was measured using CESD-10, the total score $\geq$10 was determined to present depression. Cognitive impairment was defined by the Mini-Mental State Examination (MMSE) score below 24 points.

Heart disease, stroke, chronic lung disease, asthma, liver disease, cancer, digestive disease, kidney disease, arthritis, psychiatric disease, and memory-related disease were defined based on self-report of a physician's diagnosis.

**eTable 1** Baseline characteristics in patients with and without multimorbidity

| characteristics | No multimorbidity | multimorbidity |
|---|---|---|
| **Number of patients** | 3,475 | 13,792 |
| **Male (%)** | 1,802 (51.89) | 6,429 (46.61) |
| **age** | 54.00 (48.00 – 61.00) | 58.00 (51.00 – 65.00) |
| **BMI (kg/m²)** | 23.29 (20.91 – 26.01) | 23.29 (20.91 – 26.01) |
| **SBP (mmHg)** | 121.00 (111.00 -129.00) | 132.00 (118.00 – 148.00) |
| **DBP (mmHg)** | 72.00 (66.00 – 78.00) | 77.00 (69.00 – 85.00) |
| **Pulse (times/min)** | 71.00 (65.00 – 78.00) | 72.00 (65.00 – 79.00) |
| **Waist (cm)** | 81.10 (76.00 – 87.88) | 85.00 (78.00 – 92.60) |
| **Education** | | |
| Elementary school or below | 2,083 (59.94) | 9,325 (67.61) |
| Junior high school | 785 (22.59) | 2,838 (20.58) |
| Senior high school or above | 596 (17.15) | 1626 (11.79) |
| missing | 11 (0.32) | 3 (0.02) |
| **Marital status, live alone, n (%)** | 336 (9.70) | 1,761 (12.77) |

| Characteristics | | |
|---|---|---|
| **Place of residence, n (%)** | | |
| rural | 2,005(57.70) | 8,258(59.89) |
| urban | 1,470(42.30) | 2,068(34.23) |
| **Drinking (%)** | 1,346 (38.73) | 5,287 (38.33) |
| **Smoking (%)** | 1,353 (38.94) | 5,349 (38.78) |
| **WBC (10^9)** | 5.80 (4.89 – 7.00) | 6.00 (5.00 – 7.20) |
| **MCV (fl)** | 92.38 (88.00 – 96.30) | 91.20 (86.65 – 95.43) |
| **Platelets** | 206.00 (163.00 – 249.00) | 207.00 (162.00 – 255.00) |
| **BUN (mg/dl)** | 15.10 (12.52 – 18.35) | 15.15 (12.55 – 18.21) |
| **Glucose (mg/dl)** | 100.17 (92.83 -108.18) | 102.78 (94.68 – 114.66) |
| **Creatinine (mg/dl)** | 0.75 (0.64 – 0.86) | 0.76 (0.66 – 0.88) |
| **Total Cholesterol (mmol/L)** | 4.74 (4.18 – 5.25) | 4.95 (4.33 – 5.62) |
| **Triglycerides (mmol/L)** | 0.97 (0.73 – 1.34) | 1.24 (0.88 – 1.84) |
| **HDL (mmol/l)** | 1.37 (1.18 – 1.61) | 1.25 (1.01 – 1.53) |
| **LDL (mmol/l)** | 2.86 (2.34 – 3.31) | 2.97 (2.41 – 3.59) |
| **CRP (mg/L)** | 0.79 (0.46 – 1.64) | 1.07 (0.57 – 2.24) |
| **HbA1c (%)** | 5.10 (4.80 – 5.30) | 5.10 (4.90 – 5.40) |
| **UA (mg/dl)** | 4.16 (3.47 – 5.01) | 4.34 (3.59 – 5.20) |

**eTable 2** Baseline characteristics of different multimorbidity cluster groups after data completion

| | Clusters, N= 13,792 | | | | |
|---|---|---|---|---|---|
| **Characteristics** | **respiratory and digestion** | **hypertension** | **heart and digestion** | **cancer** | **p** |
| **Age mean [SD], years** | 57.00 (50.00 - 64.00) | 60.00 (53.00 – 67.00) | 60.00 (54.00 – 67.00) | 56.00 (49.00 – 62.00) | < 0.001 |
| **SBP (mmHg)** | 121.00 (110.00 -132.00) | 145.00 (134.00 – 159.00) | 142.00 (127.00 – 154.00) | 120.00 (111.00 – 129.00) | 0.000 |
| **DBP (mmHg)** | 72.00 (65.00 -79.00) | 83.00 (75.00 – 91.00) | 81.00 (72.00 – 89.00) | 72.00 (65.00 – 78.00) | 0.000 |
| **Pulse (times/min)** | 72.00 (65.00 -79.00) | 73.00 (66.00 – 80.00) | 72.00 (65.00 – 79.00) | 72.00 (65.00 – 79.00) | < 0.001 |
| **BMI subgroups, n (%)** | | | | | < 0.001 |
| <18.5 kg/m$^2$ | 319 (10.62) | 223 (4.52) | 124 (5.55) | 209 (5.77) | < 0.001 |
| 18.5~23.9 kg/m$^2$ | 1,786 (59.43) | 2,237 (45.38) | 1,031 (46.13) | 1,970 (54.39) | < 0.001 |
| 24~28 kg/m$^2$ | 736 (24.49) | 1,608 (32.62) | 712 (31.86) | 1,056 (29.16) | < 0.001 |
| >28 kg/m$^2$ | 164 (5.46) | 862 (17.48) | 368 (16.47) | 387 (10.68) | < 0.001 |
| **Education** | | | | | < 0.001 |
| Elementary school or below | 2,110 | 3,301 | 1,660 | 2,256 | < 0.001 |
| Junior high school | 581 | 1,008 | 373 | 876 | < 0.001 |
| Senior high school or above | 314 | 621 | 202 | 489 | < 0.001 |
| **WBC (10^9/L)** | 5.90 | 6.10 | 6.00 | 5.90 | < 0.001 |

| | | | | | |
|---|---|---|---|---|---|
| | (4.90 -7.10) | (5.00 – 7.40) | (5.00 – 7.20) | (4.90 – 7.10) | |
| MCV (fl) | 91.54 | 91.20 | 90.90 | 90.70 | < 0.001 |
| | (86.70 -96.00) | (87.00 – 95.50) | (86.40 – 95.40) | (85.70 – 95.00) | |
| Platelets (10^9/L) | 202.00 | 205.00 | 209.00 | 209.00 | < 0.001 |
| | (158.00 -248.00) | (162.00 – 255.00) | (165.00 – 260.00) | (164.00 – 259.00) | |
| BUN (mg/dl) | 15.15 | 15.18 | 15.29 | 15.07 | 0.153 |
| | (12.44 -18.26) | (12.60 – 18.29) | (12.66 – 18.35) | (12.49 – 17.98) | |
| Glucose (mmol/L) | 5.59 | 5.80 | 5.80 | 5.64 | < 0.001 |
| | (5.18 -6.14) | (5.34 – 6.53) | (5.29 – 6.49) | (5.22 – 6.24) | |
| Creatinine (mg/dl) | 0.76 | 0.78 | 0.76 | 0.75 | < 0.001 |
| | (0.64 – 0.88) | (0.67 – 0.92) | (0.66 – 0.88) | (0.63 – 0.87) | |
| Total Cholesterol (mmol/L) | 4.81 | 4.99 | 5.06 | 4.89 | < 0.001 |
| | (4.24 – 5.46) | (4.39 – 5.69) | (4.41 – 5.72) | (4.30 – 5.59) | |
| Triglycerides (mmol/L) | 1.13 | 1.33 | 1.40 | 1.28 | < 0.001 |
| | (0.82 – 1.65) | (0.92 – 2.00) | (0.96 – 2.05) | (0.89 – 1.88) | |
| HDL (mmol/l) | 1.34 | 1.22 | 1.22 | 1.23 | < 0.001 |
| | (1.09 – 1.60) | (0.99 – 1.50) | (1.00 – 1.50) | (0.98 – 1.51) | |
| LDL (mmol/l) | 2.86 | 2.98 | 3.02 | 2.94 | < 0.001 |
| | (2.35 – 3.46) | (2.44 – 3.65) | (2.43 – 3.65) | (2.37 – 3.58) | |
| CRP (mg/L) | 0.92 | 1.24 | 1.15 | 0.99 | < 0.001 |
| | (0.52 – 2.11) | (0.63 – 2.55) | (0.61 – 2.50) | (0.53 – 2.04) | |
| HbA1c (%) | 5.10 | 5.20 | 5.20 | 5.10 | < 0.001 |
| | (4.90 – 5.40) | (4.90 – 5.50) | (4.90 – 5.50) | (4.90 – 5.40) | |
| UA (mg/dl) | 4.13 | 4.53 | 4.31 | 4.26 | < 0.001 |
| | (3.46 – 4.97) | (3.76 – 5.44) | (3.57 – 5.19) | (3.56 – 5.09) | |

**eTable 3** Baseline characteristics of the clusters with multimorbidity from different clusters

| | Clusters, N= 13,792 | | | | |
|---|---|---|---|---|---|
| Characteristics | respiratory and digestion | hypertension | heart and digestion | cancer | *p* |
| Numbers (n) | 3,005 (21.79) | 4,930 (35.75) | 2,235 (16.21) | 3,622 (26.26) | < 0.001 |
| Male (%) | 1,483 (49.35) | 2,414 (48.97) | 849 (37.99) | 1,683 (46.47) | < 0.001 |
| Age mean [SD], years | 57.00 | 60.00 | 60.00 | 56.00 | < 0.001 |
| | (50.00 - 64.00) | (53.00 – 67.00) | (54.00 – 67.00) | (49.00 – 62.00) | |
| SBP (mmHg) | 119.00 | 146.00 | 143.00 | 119.00 | 0.000 |
| | (110.00 -129.00) | (136.00 – 160.00) | (129.00 – 156.00) | (111.00 – 128.00) | |
| DBP (mmHg) | 70.00 | 84.00 | 82.00 | 71.00 | 0.000 |
| | (64.00 -77.00) | (76.00 – 92.00) | (74.00 – 90.00) | (65.00 – 77.00) | |
| Pulse (times/min) | 72.00 | 73.00 | 72.00 | 72.00 | 0.004 |
| | (65.00 -79.00) | (66.00 – 80.00) | (65.00 – 79.00) | (65.00 – 79.00) | |
| BMI subgroups, n (%) | | | | | < 0.001 |
| <18.5 kg/m$^2$ | 281 (9.35) | 196 (3.98) | 102 (4.56) | 172 (4.75) | < 0.001 |
| 18.5~23.9 kg/m$^2$ | 1,440 (47.92) | 1,895 (38.44) | 852 (38.12) | 1,609 (44.42) | < 0.001 |
| 24~28 kg/m$^2$ | 565 (18.80) | 1,415 (28.70) | 601 (26.89) | 795 (21.95) | < 0.001 |

| | | | | | |
|---|---|---|---|---|---|
| >28 kg/m$^2$ | 98 (3.26) | 765 (15.52) | 322 (14.41) | 275 (7.59) | < 0.001 |
| missing | 621 (20.67) | 659 (13.37) | 358 (16.02) | 771 (21.29) | |
| **Education** | | | | | < 0.001 |
| Elementary school or below | 2,109 | 3,300 | 1,660 | 2,256 | < 0.001 |
| Junior high school | 581 | 1008 | 373 | 876 | < 0.001 |
| Senior high school or above | 314 | 621 | 202 | 489 | < 0.001 |
| missing | 1 | 1 | 0 | 1 | |
| **Marital status, live alone, n (%)** | 315 (10.49) | 749 (15.19) | 364 (16.29) | 333 (9.19) | < 0.001 |
| **Place of residence, n (%)** | | | | | |
| rural | 1,993 | 2,765 | 1,363 | 2,137 | < 0.001 |
| urban | 1,012 | 2,165 | 872 | 1,485 | < 0.001 |
| **WBC (10^9/L)** | 5.80 (4.84 -7.10) | 6.10 (5.00 – 7.40) | 6.00 (5.00 – 7.27) | 5.90 (4.90 – 7.13) | < 0.001 |
| **MCV (fl)** | 91.72 (86.90 -96.00) | 91.20 (87.00 – 95.40) | 91.00 (86.80 – 95.30) | 91.00 (86.00 – 95.00) | < 0.001 |
| **Platelets (10^9/L)** | 201.00 (157.00 -247.00) | 206.00 (163.00 – 256.00) | 210.00 (166.00 – 262.00) | 209 (163.00 – 258.00) | < 0.001 |
| **BUN (mg/dl)** | 15.15 (12.38 -18.23) | 15.17 (12.67 – 18.26) | 15.27 (12.72 – 18.35) | 15.04 (12.46 – 18.01) | 0.116 |
| **Glucose (mmol/L)** | 5.58 (5.16 -6.10) | 5.82 (5.35 – 6.56) | 5.81 (5.30 – 6.52) | 5.64 (5.21 – 6.21) | < 0.001 |
| **Creatinine (mg/dl)** | 0.75 (0.64 – 0.87) | 0.78 (0.67 – 0.93) | 0.75 (0.66 – 0.88) | 0.75 (0.64 – 0.86) | < 0.001 |
| **Total Cholesterol (mmol/L)** | 4.81 (4.23 – 5.43) | 5.00 (4.40 – 5.72) | 5.08 (4.40 – 5.77) | 4.91 (4.29 – 5.59) | < 0.001 |
| **Triglycerides (mmol/L)** | 1.08 (0.79 – 1.52) | 1.33 (0.92 – 1.95) | 1.39 (0.97 – 2.06) | 1.23 (0.87 – 1.80) | < 0.001 |
| **HDL (mmol/l)** | 1.34 (1.11 – 1.61) | 1.22 (0.99 – 1.49) | 1.22 (0.99 – 1.50) | 1.24 (0.98 – 1.51) | < 0.001 |
| **LDL (mmol/l)** | 2.87 (2.36 – 3.46) | 3.00 (2.45 – 3.65) | 3.03 (2.44 – 3.70) | 2.96 (2.38 – 3.58) | < 0.001 |
| **CRP (mg/L)** | 0.86 (0.49 – 1.96) | 1.24 (0.65 – 2.51) | 1.15 (0.62 – 2.50) | 0.96 (0.52 – 1.97) | < 0.001 |
| **HbA1c (%)** | 5.10 (4.90 – 5.40) | 5.20 (4.90 – 5.50) | 5.20 (4.90 – 5.50) | 5.10 (4.90 – 5.40) | < 0.001 |
| **UA (mg/dl)** | 4.08 (3.43 – 4.90) | 4.54 (3.74 – 5.46) | 4.31 (3.56 – 5.18) | 4.25 (3.55 – 5.09) | < 0.001 |
| **Smoking history, n (%)** | 1270 (42.26) | 1916 (38.86) | 786 (35.17) | 1377 (38.02) | < 0.001 |
| missing | 0 | 5 | 4 | 6 | |
| **Drinking history, n (%)** | 1197 (39.83) | 1964 (39.84) | 772 (34.54) | 1354 (37.38) | < 0.001 |
| missing | 2 | 9 | 5 | 7 | |

SBP, systolic blood pressure; DBP, diastolic blood pressure; BMI body mass index, WBC, white blood cell, MCV, mean corpuscular volume, BUN, Blood Urea Nitrogen;

HDL, high density lipoprotein; LDL, low density lipoprotein; CRP, C-reactive protein; HbA1c, Glycated hemoglobin; UA, uric acid.

**eTable 4** summary of mortality rates for each disease

| Prevalence condition | 2-year mortality (%) | 4-year mortality (%) | 7-year mortality (%) |
|---|---|---|---|
| Hypertension | 4.37 | 10.86 | 15.73 |
| Diabetes | 4.95 | 12.12 | 17.52 |
| Cancer | 5.35 | 14.44 | 17.65 |
| Lung disease | 5.77 | 15.64 | 21.35 |
| Heart problem | 3.85 | 11.03 | 16.17 |
| Stroke | 9.20 | 22.80 | 29.60 |
| Psychiatric problem | 5.13 | 13.55 | 19.05 |
| Arthritis | 2.94 | 8.24 | 12.49 |
| Dyslipidemia | 2.89 | 7.76 | 11.51 |
| Liver disease | 4.12 | 10.29 | 13.38 |
| Kidney disease | 3.58 | 9.95 | 14.16 |
| Stomach disease | 3.05 | 7.49 | 11.05 |
| Asthma | 6.37 | 16.07 | 25.35 |
| Memory problem | 12.07 | 26.93 | 37.15 |
| Depression | 3.68 | 9.57 | 13.63 |
| Cognition impairment | 2.68 | 7.10 | 10.51 |
| Adiposity | 1.64 | 5.14 | 7.74 |
| Hyperuricemia | 6.73 | 15.55 | 20.88 |
| Anaemia | 4.37 | 12.32 | 16.69 |

**Table S5** Association Between stratification of clustered populations and multimorbidity

| Hazard ratio (95% CI) | cancer | respiratory and digestion | hypertension | heart and digestion |
|---|---|---|---|---|
| **Sex** | | | | |
| male | Ref. | 1.274 (1.108, 1.465) ** | 1.335 (1.177, 1.514) ** | 1.265 (1.075, 1.489) ** |
| female | Ref. | 1.226 (1.053, 1.428) ** | 1.374 (1.202, 1.570) ** | 1.297 (1.111, 1.514) ** |
| **Age** | | | | |
| < 60 y | Ref. | 1.170(1.010, 1.355) * | 1.135 (0.990, 1.302) | 1.026 (0.858, 1.226) |
| > 60 y | Ref. | 1.251 (1.078, 1.452) ** | 1.290 (1.132, 1.469) ** | 1.198 (1.029, 1.395) * |
| **Medication use** | | | | |
| Yes | Ref. | 1.427 (1.219, 1.671) ** | 1.269 (1.117, 1.442) ** | 1.195 (0.954, 1.496) |
| No | Ref. | 1.021 (0.880, 1.185) | 1.260 (1.096, 1.448) ** | 1.062 (0.913,1.236) |
| | | | | *p < 0.05, **P < 0.01 |

**Table S6** Association between of multimorbidity clusters and patient demographics. Coefficients are reported in odds ratios (for logit models) with 95% confidence intervals.

| Predictors | Cluster 1 OR (95% CI) | Cluster 2 OR (95% CI) | Cluster 3 OR (95% CI) | Cluster 4 OR (95% CI) |
|---|---|---|---|---|
| Gender | | | | |
| Male | 1 | 1 | 1 | 1 |
| Female | 0.541** | 0.882 | 0.676 | 0.522** |

| | | | | |
|---|---|---|---|---|
| | (0.374 – 0.784) | (0·687 – 1.133) | (0.457 – 1.001) | (0.346 – 0.789) |
| Smoking status | | | | |
| Non-smoker | 1 | 1 | 1 | 1 |
| smoker | 1.317 | 1.581** | 1.350 | 1.215 |
| | (0.929 – 1.869) | (1.247 – 2.004) | (0.923 – 1.975) | (0.842 – 1.754) |
| No-drinking | 1 | 1 | 1 | 1 |
| drinking | 0.998 | 1.027 | 0.956 | 1.133 |
| | (0.748 – 1.331) | (0.831– 1.271) | (0.679 – 1.346) | (0.820 – 1.565) |
| BMI < 24 | 1 | 1 | 1.00 | 1.00 |
| BMI ≥ 24 | 0.616 | 0.617** | 0.662** | 0.710* |
| | (0.442 – 0.860) | (0.515 – 0.740) | (0.491 – 0.892) | (0.506– 0.997) |
| Age < 60 | 1.00 | 1.00 | 1.00 | 1·00 |
| Age ≥ 60 | 4.703** | 4.261** | 6.115** | 4.703** |
| | （3.468 – 6.377） | (3.475 – 5.224) | (4.234 -8.832) | (3.468 – 6.377) |

*p < 0.05 **p <0.01

**Table S7 Association between of multimorbidity clusters and service utilisation and mortality.** Coefficients are reported in incidence rate ratios (for negative binomial models) and odds ratios (for logit models) with 95% confidence intervals.

| Predictors | hospital nights last recent time (2011) IRR (95% CI) | 2-year mortality OR (95% CI) | 4-year mortality OR (95% CI) | 7-year mortality OR (95% CI) |
|---|---|---|---|---|
| cancer | 1 | 1 | 1 | 1 |
| respiratory and digestion | 1.386** | 1.329 | 1.527** | 1.589** |
| | (1.117 – 1.721) | (0.986 – 1.791) | (1.264 – 1.846) | (1.354 – 1.864) |
| hypertension | 1.265* | 1.962** | 1.963** | 2.028** |
| | (1.040 – 1.540) | (1.522 – 2.529) | (1.662 – 2.318) | (1.761 – 2.335) |
| heart and digestion | 1.755** | 1..505* | 1.667** | 1.737** |
| | (1.431 – 2.152) | (1.101 – 2.056) | (1.365 – 2.036) | (1.466 – 2.057) |
| Male | 1 | 1 | 1 | 1 |
| female | 0.685** | 0.968 | 1.453** | 0.699** |
| | (0.536 – 0.875) | (0.755 – 1.242) | (1.195 – 1.765) | (0.592 – 0.825) |
| BMI < 24 | 1 | 1 | 1 | 1 |
| BMI ≥ 24 | 1.246** | 0.583** | 0.661** | 0.637** |
| | (1.057 – 1.468) | (0.458 – 0.743) | (0.567 – 0.770) | (0.560 – 0.725) |
| Age < 60 | 1 | 1 | 1 | 1 |
| Age ≥ 60 | 1.464** | 3.865** | 4.631** | 4.807** |
| | (1.206 – 1.776) | (3.005 – 4.971) | (3.937 – 5.448) | (4.200 – 5.501) |
| No smoking | 1 | 1 | 1 | 1 |
| Smoking | 0.884 | 1.556** | 1.405** | 1.416** |
| | (0.700 – 1.117) | (1.167 – 2.075) | (1.169 – 1.689) | (1.210 – 1.657) |
| No drinking | 1 | 1 | 1 | 1 |
| Drinking | 0.954 | 0.968 | 0.975 | 1.022 |
| | (0.783– 1.161) | (0.755 – 1.242) | (0.830 – 1.146) | (0.890 – 1.173) |

**Figure S1** Flowchart of Study Participant Selection

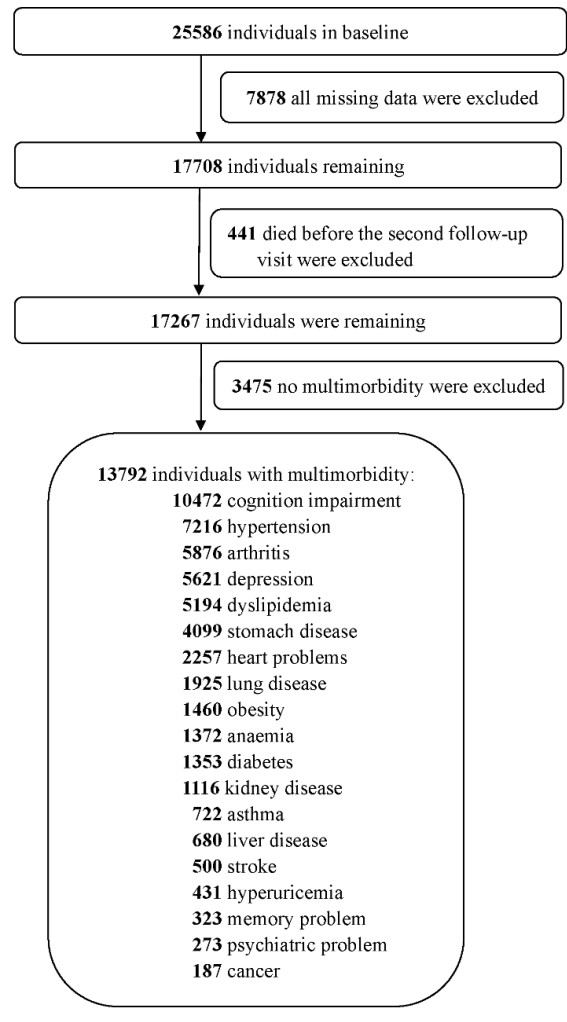

**Figure S2** Diagrams depicting reduction and clustering of data dimensions

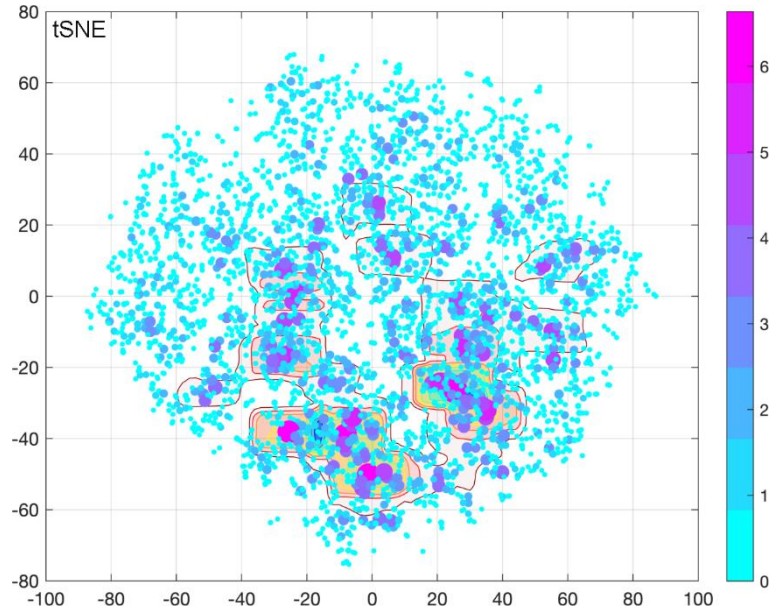

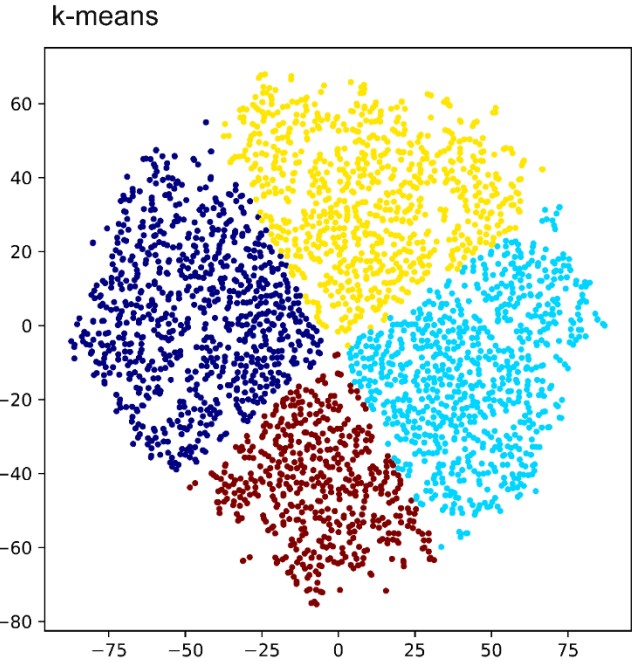

**Figure S3** The appropriate number of clustering groups for the 3 to 8 class clusters by different clustering evaluation indicators.

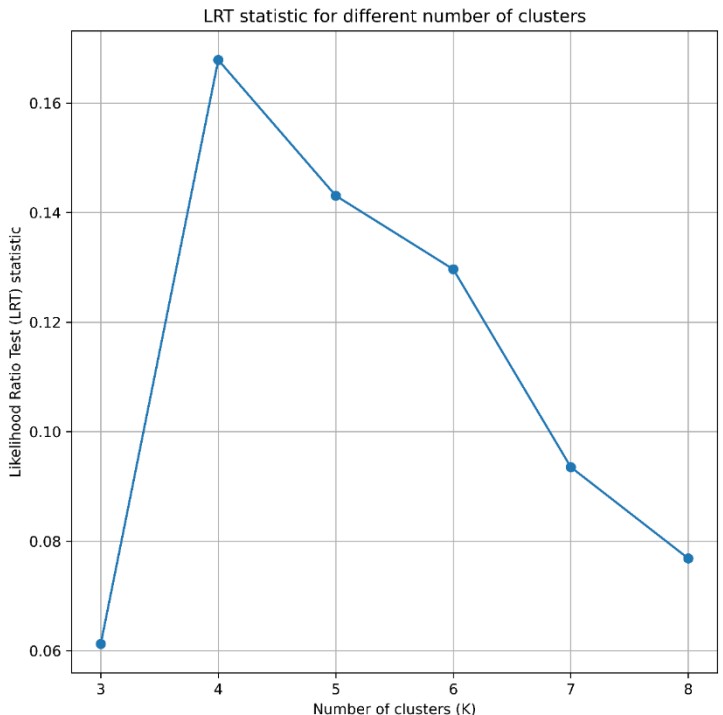

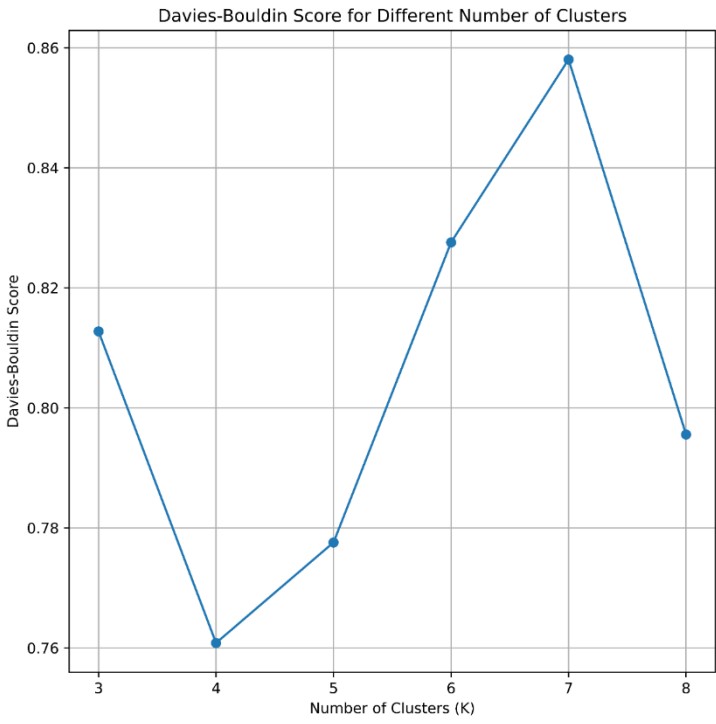

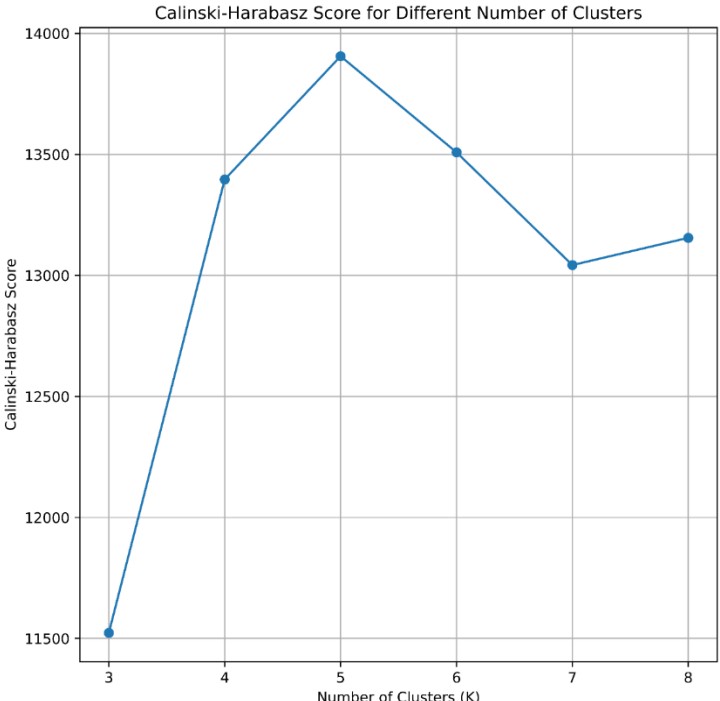

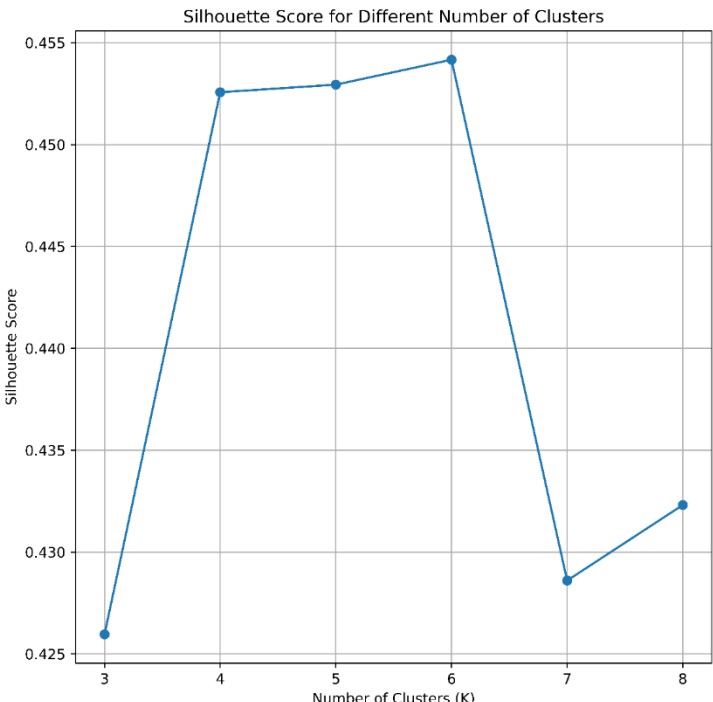

**Figure S4** Identification of prominent condition(s) by cluster category. O/E ratio: Observation/expected ratio; Exclusivity: intra-cluster prevalence of comorbidities

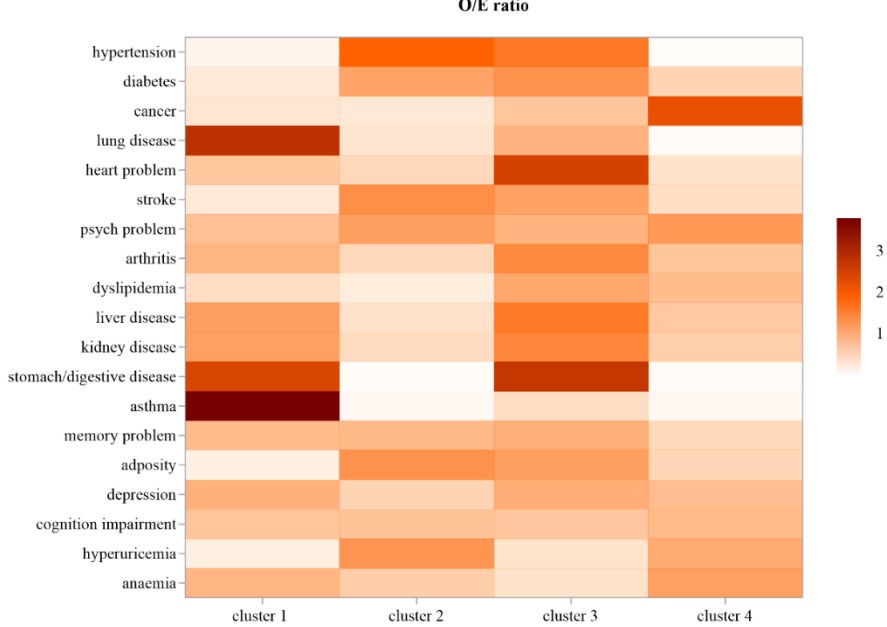

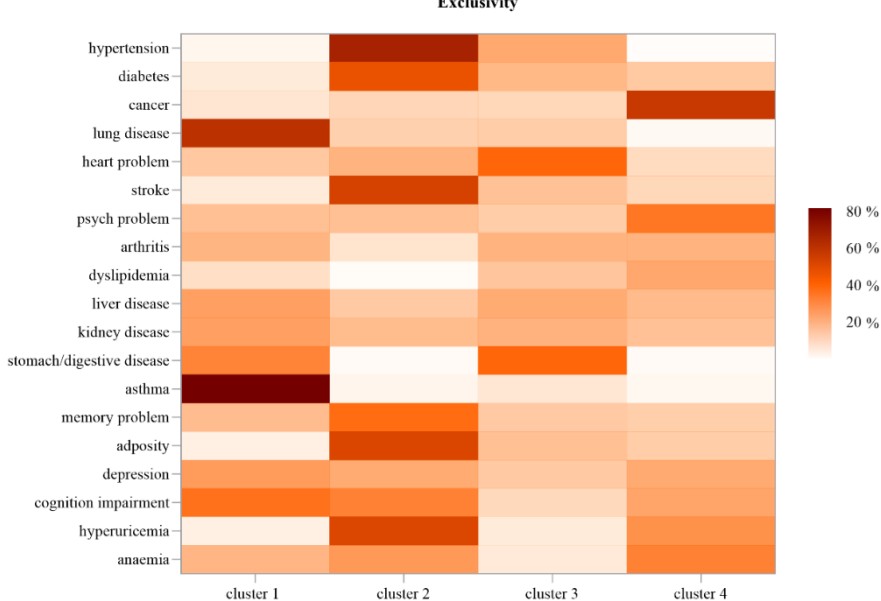

**Figure S5** Kaplan-Meier survival curves based on multimorbidity clusters

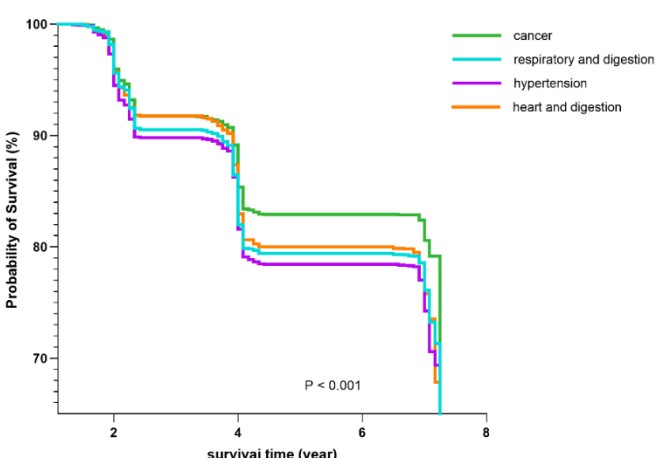

**Figure S6** Correlation of pairwise combinations of different disease

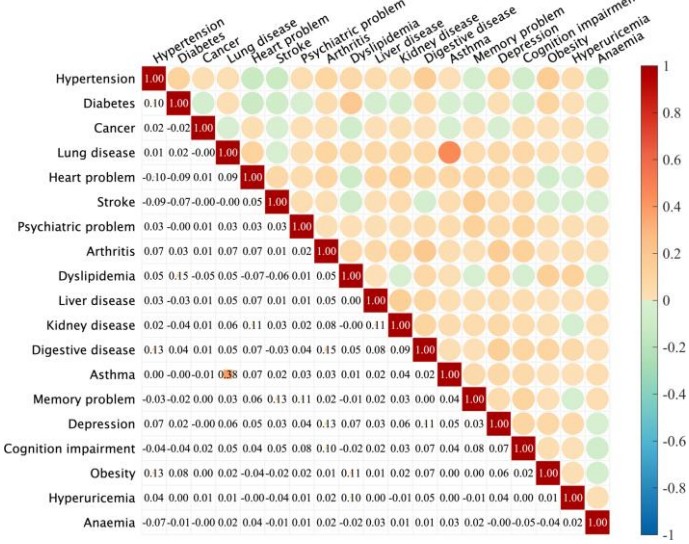

**Figure S7 Health care burden for each cluster in 2011,2013, 2015, and 2018.** A represent hospital nights last recent time; B represent hospital stay times a year; C represent hospitalization out-of-pocket expenditure a year (yuan); D represent hospitalization total expenditure a year (yuan).

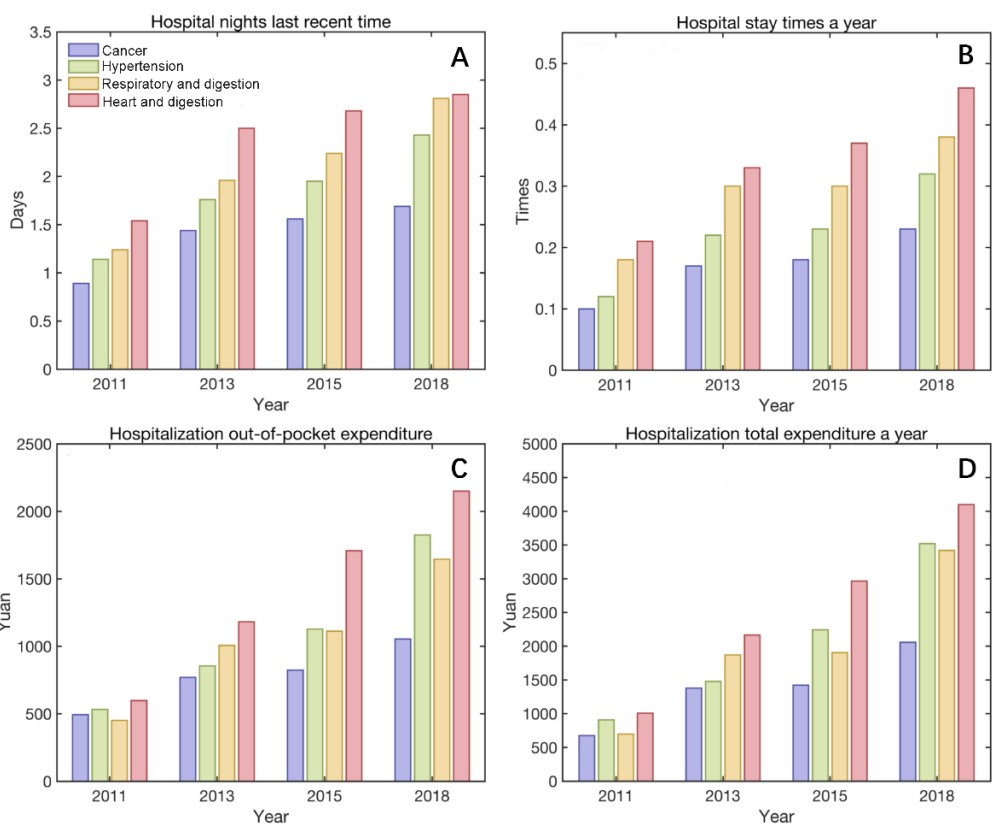

# Alternative approach

**Alternative analysis**

To assess the stability of clustering results, we also used alternative methods to divide individuals into four clusters (Figure S8, Figure S9). This surrogate analysis also partitioned individuals into four clusters, which was more consistence with the primary analysis (Table S8). In terms of disease burden and mortality, the unspecific group experienced the most, contrast to the lung and asthma group, which had the least both (Figure S10, Table S9). The stratified analysis revealed that the lung and asthma group faced greater risk compared to other groups, with each showing statistical significance (Figure S11).

**Table S8** Baseline characteristics of the clusters with multimorbidity from different clusters

| Characteristics | Clusters, N= 13,792 | | | | |
| | Cluster 1 hypertension | Cluster 2 unspecific | Cluster 3 Lung and asthma | Cluster 4 psychiatric and digestion | *p* |
| --- | --- | --- | --- | --- | --- |
| **Numbers (n)** | 6,163 (44.69) | 3,535 (25.63) | 1,894 (13.73) | 2,200 (15.95) | 0.000 |
| **Male (%)** | 2,815 (45.68) | 1,636 (46.28) | 1,051 (55.49) | 927 (42.14) | < 0.001 |
| **Age mean [SD], years** | 59.00 | 55.00 | 61.00 | 55.00 | < 0.001 |

| | | | | | |
|---|---|---|---|---|---|
| | (53.00 - 67.00) | (49.00 – 62.00) | (55.00 – 69.00) | (48.00 – 61.00) | |
| **SBP (mmHg)** | 146.00 | 119.00 | 134.00 | 117.00 | 0.000 |
| | (136.00 -159.00) | (111.00 – 127.00) | (120.00 – 149.00) | (109.00 – 125.00) | |
| **DBP (mmHg)** | 84.00 | 71.00 | 77.00 | 70.00 | 0.000 |
| | (76.00 -92.00) | (65.00 – 77.00) | (69.00 – 86.00) | (63.00 – 76.00) | |
| **Pulse (times/min)** | 72.00 | 72.00 | 74.00 | 71.00 | < 0.001 |
| | (65.00 -80.00) | (65.00 – 78.00) | (67.00 – 81.00) | (65.00 – 78.00) | |
| **BMI subgroups, n (%)** | | | | | < 0.001 |
| <18.5 kg/m$^2$ | 236 (3.83) | 170 (4.81) | 176 (9.29) | 169 (7.68) | < 0.001 |
| 18.5~23.9 kg/m$^2$ | 2,368 (38.42) | 1,543 (43.65) | 803 (42.40) | 1,082 (49.18) | < 0.001 |
| 24~28 kg/m$^2$ | 1,798 (29.17) | 763 (21.58) | 389 (20.54) | 426 (19.36) | 0.000 |
| >28 kg/m$^2$ | 913 (14.81) | 315 (8.91) | 168 (8.87) | 62 (2.82) | < 0.001 |
| missing | 848 (13.76) | 744 (21.05) | 358 (18.90) | 461 (20.95) | |
| **Education** | | | | | < 0.001 |
| Elementary school or below | 4,202 | 2,185 | 1,424 | 1,514 | 0.000 |
| Junior high school | 1,217 | 858 | 317 | 446 | < 0.001 |
| Senior high school or above | 743 | 491 | 152 | 240 | < 0.001 |
| missing | 1 | 1 | 1 | 0 | |
| **Marital status, live alone, n (%)** | 315 (10.49) | 749 (15.19) | 364 (16.29) | 333 (9.19) | < 0.001 |
| **Place of residence, n** | | | | | |
| rural | 3,491 | 2,079 | 1,239 | 1,449 | 0.000 |
| urban | 2,672 | 1,456 | 655 | 751 | 0.000 |
| **WBC (10^9/L)** | 6.10 | 5.90 | 6.10 | 5.70 | < 0.001 |
| | (5.00 -7.30) | (4.90 – 7.11) | (5.10 – 7.60) | (4.77 – 6.90) | |
| **MCV (fl)** | 91.20 | 90.90 | 92.00 | 91.00 | < 0.001 |
| | (87.00 -95.30) | (86.00 – 95.00) | (87.60 – 96.40) | (86.40 – 95.40) | |
| **Platelets (10^9/L)** | 207.00 | 209.00 | 206.00 | 202.00 | < 0.001 |
| | (163.50 -258.00) | (164.00 – 259.00) | (160.00 – 256.00) | (157.00 – 246.00) | |
| **BUN (mg/dl)** | 15.15 | 15.01 | 15.43 | 14.96 | 0.004 |
| | (12.63 -18.26) | (12.46 – 17.99) | (12.80 – 18.57) | (12.35 – 18.15) | |
| **Glucose (mmol/L)** | 5.82 | 5.65 | 5.65 | 5.58 | < 0.001 |
| | (5.35 -6.56) | (5.21 – 6.23) | (5.24 – 6.26) | (5.16 – 6.09) | |
| **Creatinine (mg/dl)** | 0.77 | 0.75 | 0.78 | 0.73 | < 0.001 |
| | (0.67 – 0.90) | (0.63 – 0.86) | (0.67 – 0.90) | (0.63 – 0.86) | |
| **Total Cholesterol (mmol/L)** | 5.01 | 4.92 | 4.94 | 4.80 | < 0.001 |
| | (4.40 – 5.72) | (4.29 – 5.61) | (4.30 – 5.62) | (4.24 – 5.43) | |
| **Triglycerides (mmol/L)** | 1.34 | 1.25 | 1.19 | 1.07 | < 0.001 |
| | (0.93 – 1.34) | (0.88 – 1.83) | (0.85 – 1.77) | (0.79 – 1.53) | |
| **HDL (mmol/l)** | 1.21 | 1.23 | 1.29 | 1.32 | < 0.001 |
| | (1.00 – 1.49) | (0.98 – 1.50) | (1.06 – 1.60) | (1.09 – 1.60) | |
| **LDL (mmol/l)** | 3.01 | 2.97 | 2.91 | 2.89 | < 0.001 |
| | (2.45 – 3.67) | (2.39 – 3.58) | (2.36 – 3.55) | (2.36 – 3.46) | |
| **CRP (mg/L)** | 1.21 | 0.97 | 1.22 | 0.78 | < 0.001 |
| | (0.64 – 2.48) | (0.52 – 1.99) | (0.63 – 2.79) | (0.47 – 1.65) | |

| | | | | | |
|---|---|---|---|---|---|
| **HbA1c (%)** | 5.20 | 5.10 | 5.10 | 5.10 | < 0.001 |
| | (4.90 – 5.50) | (4.90 – 5.40) | (4.90 – 5.40) | (4.80 – 5.40) | |
| **UA (mg/dl)** | 4.48 | 4.26 | 4.43 | 4.03 | < 0.001 |
| | (3.68 – 5.39) | (3.55 – 5.08) | (3.65 – 5.34) | (3.40 – 4.77) | |
| **Smoking history, n (%)** | 2288 (37.12) | 1337 (37.82) | 935 (49.37) | 789 (35.86) | < 0.001 |
| missing | 8 | 6 | 0 | 1 | |
| **Drinking history, n (%)** | 2342 (38.00) | 1320 (37.34) | 825 (43.56) | 800 (36.36) | < 0.001 |
| missing | 1 | 7 | 0 | 3 | |

**Table S9** Association of different group with mortality risk in 7-year follow-ups

| Hazard ratio (95% CI) | Cluster 2 | Cluster 1 | Cluster 3 | Cluster 4 |
|---|---|---|---|---|
| | unspecific | hypertension | Lung and asthma | psychiatric and digestion |
| | | **Cluster 1 vs cluster 2** | **Cluster 1 vs cluster 3** | **Cluster 1 vs cluster 4** |
| **Unadjusted HR (95% CI)** | Ref. | 1.331 (1.216, 1.457) ** | 1.745 (1.565, 1.947) ** | 1.050 (0.930, 1.184) |
| **Model 1[a]** | Ref. | 1.115 (1.017, 1.223) * | 1.372 (1.226, 1.534) ** | 1.072 (0.949, 1.210) |
| **Model 2[b]** | Ref. | 1.205 (1.076, 1.349) * | 1.432 (1.248, 1.643) ** | 1.083 (0.932, 1.258) |
| **Unadjusted HR[c] (95% CI)** | Ref. | 1.796 (1.464, 2.202) ** | 2.621 (2.065, 3.327) ** | 1.027 (0.763, 1.383) |
| **Model 1[ac]** | Ref. | 1.382 (1.123, 1.701) * | 1.622 (1.271, 2.069) ** | 1.161 (0.860, 1.568) |
| **Model 2[bc]** | Ref. | 1.333 (1.077, 1.649) * | 1.464 (1.136, 1.888) * | 1.083 (0.797, 1,473) |
| **Model 1[ad]** | Ref. | 1.131 (1.032, 1.240) * | 1.385 (1.240, 1.547) ** | 1.076 (0.954, 1.213) |
| **Model 2[bd]** | Ref. | 1.102 (1.003, 1.211) * | 1.359 (1.210, 1.525) ** | 1.040 (0.919, 1.177) |
| **Death, n (%)** | 286 (8.09) | 881 (14.29) | 404 (21.33) | 172 (7.82) |

Abbreviations: CI confidence interval, RR rate ratio, ref. reference. * p < 0.05 ** p < 0.01

[a] Adjusted for age, sex.

[b] Adjusted for age, sex, smoking ever, drinking ever, BMI, medication use and live in rural or urban. RD, risk difference;

[c] Sensitivity analysis 1: Delete the missing diseases. Specifically, there were deficiencies in hypertension, diabetes, cancer, lung disease, heart problem, stroke, psych problem, arthritis, dyslipidemia, liver disease, kidney disease, stomach/digestive disease, asthma, memory problem, depression, cognition impairment, hyperuricemia, obesity and anaemia;

[d] Sensitivity analysis 2: Multiple imputation of data.

**Figure S8**     The appropriate number of clustering groups for the 3 to 8 class clusters by different clustering evaluation indicators

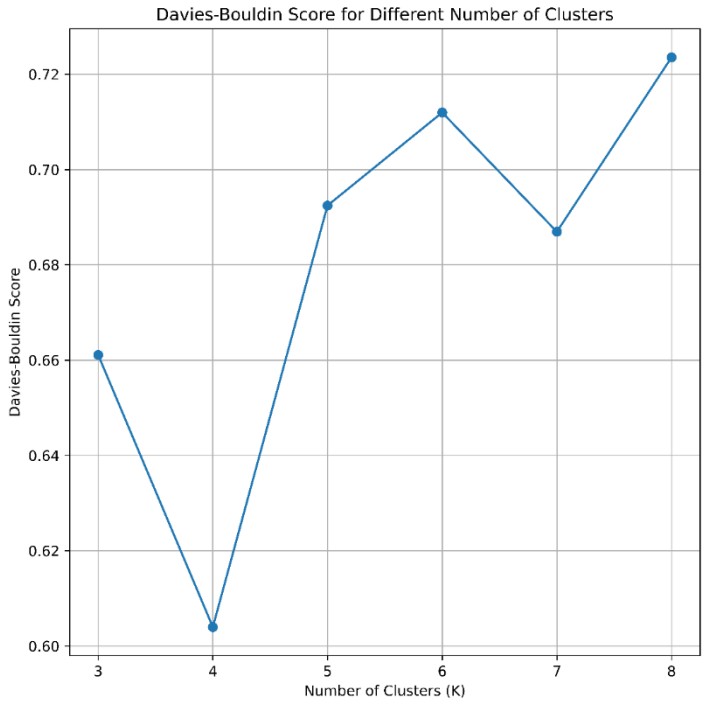

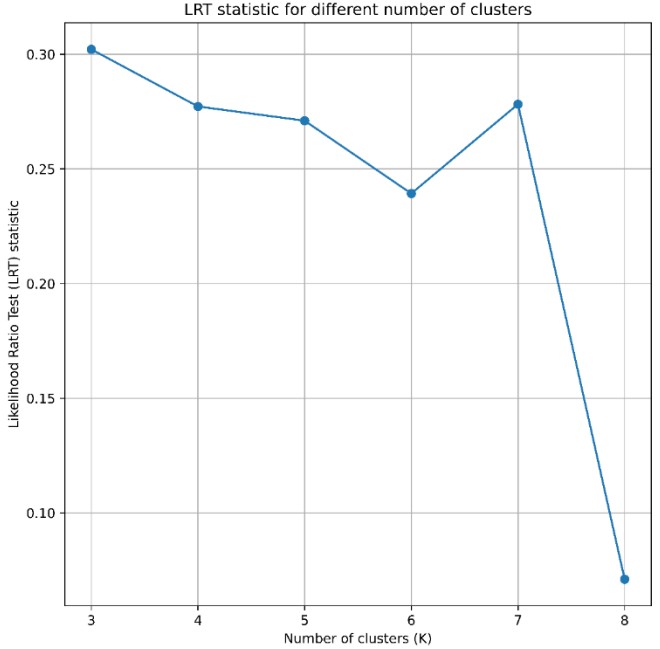

**Figure S9** Identification of prominent condition(s) by cluster category

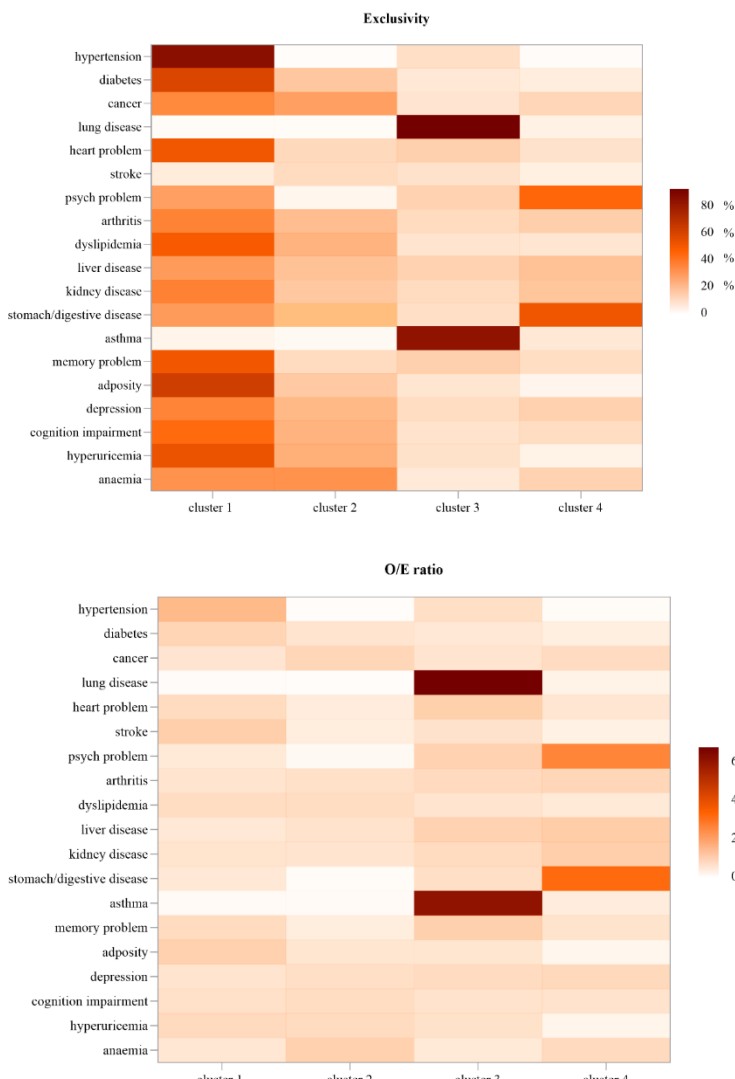

**Figure S10 Health care burden for each cluster in 2011,2013, 2015, and 2018.** A represent hospital nights last recent time; B represent hospital stay times a year; C represent hospitalization out-of-pocket expenditure a year (yuan); D represent hospitalization total expenditure a year (yuan).

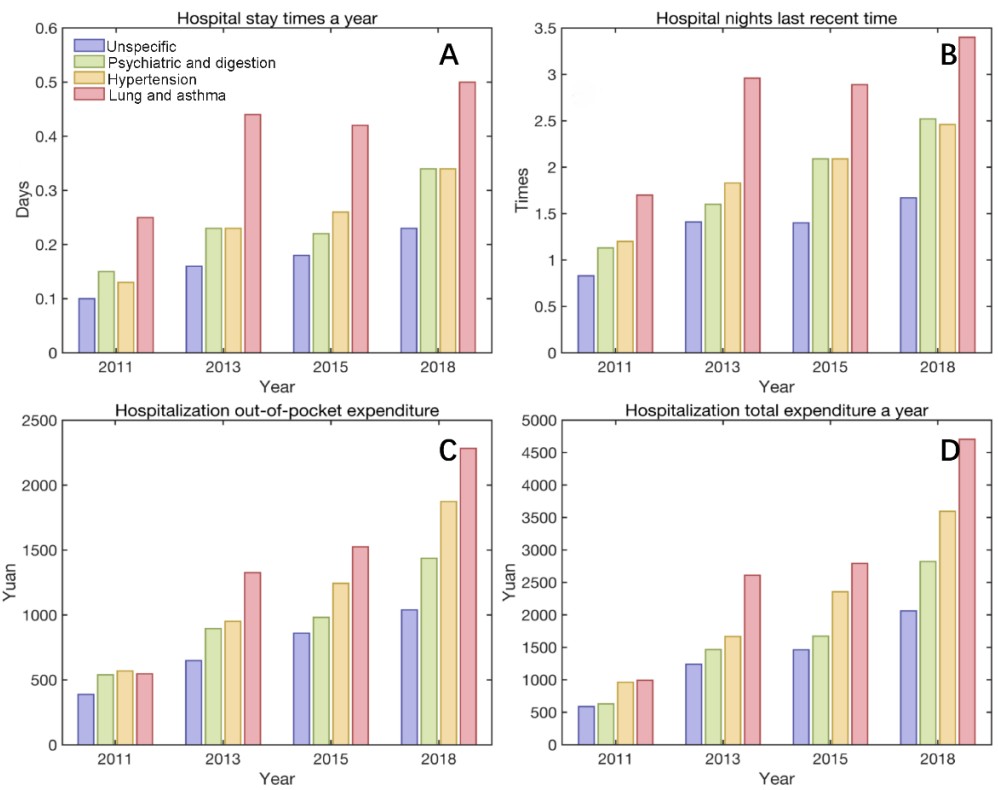

**Figure S11** Association Between stratification of clustered populations and multimorbidity

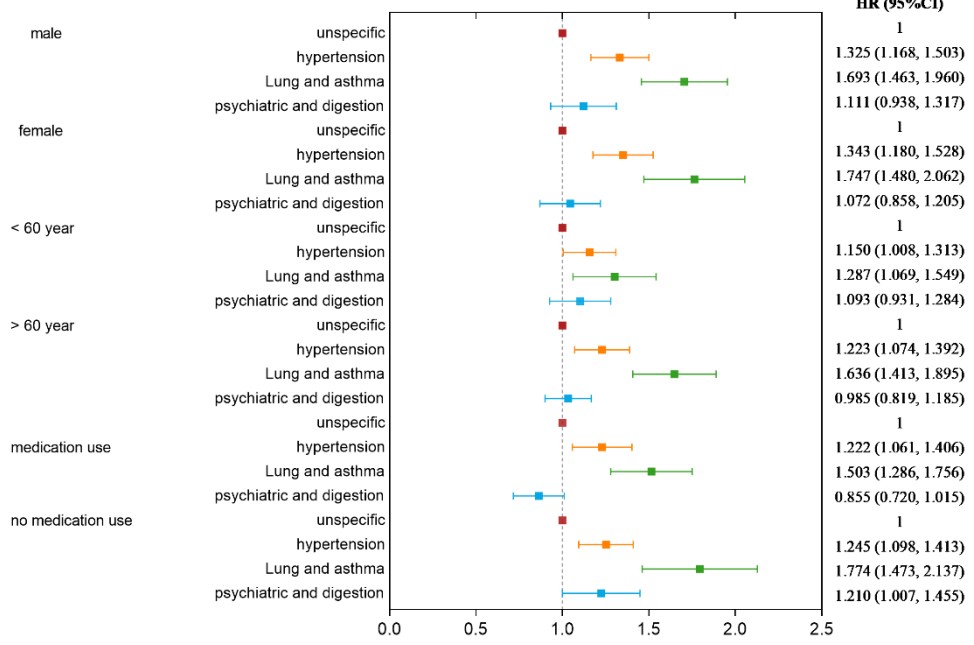