# OpenReview forum: "Cluster analysis of multimorbidity and healthcare burden based on machine learning: results from CHARLS"
_KDD.org/2024/Workshop/AIDSH — KDD-AIDSH 2024 Poster_

### Official Review · Reviewer_jds4 · 2024-06-12

**Rating:** 4
**Confidence:** 3

**Review:**

The study addresses the global issue of multimorbidity, especially in the context of an aging population. It highlights the increased risk of early death, extended hospital stays, and higher healthcare expenses faced by individuals with multiple chronic diseases. The study applied T-distributed Stochastic Neighbor Embedding (tSNE) for dimensionality reduction and k-means clustering to identify distinct multimorbidity patterns. Four distinct clusters were identified, each with unique healthcare utilization and mortality rates. The clusters were characterized by different combinations of diseases such as cancer, respiratory and digestive issues, hypertension, and heart and digestive problems. The study found varying risks of healthcare burdens and death among the clusters. It also revealed correlations among various diseases and their impact on mortality risk.

However, there are still some shortcomings:
1. While the application of tSNE and k-means clustering to multimorbidity is a valid approach, the paper does not clearly articulate the novelty of the methodological approach compared to existing literature. The use of these techniques is relatively common in the field of machine learning for similar purposes.
2. The typesetting should be improved. It is hard to read, especially Page 2.
3. The paper does not detail the robustness of the clustering results to variations in the data or the sensitivity to the choice of parameters in the clustering algorithms. More rigorous validation, such as cross-validation or external validation on a separate dataset, could strengthen the findings.

---

### Official Review · Reviewer_zj7Z · 2024-06-13
**Review comments**

**Rating:** 3
**Confidence:** 5

**Review:**

In this work, the authors conduct a statistical cluster analysis of multimorbidity and healthcare burden using the CHARLS dataset. The research question may be clinically meaningful, but the methodology lacks novelty, making this work less suitable for the KDD audience. The paper also has numerous presentation issues. My major comments are:

1. This paper has no references and fails to adhere to the formatting instructions. There are many formatting issues, such as image placement and spelling errors (e.g., "condiction" should be "condition," "Harabase" should be "Harabasz").
2. The proposed method pipeline uses a general statistical approach, including KMeans, UMAP, and t-SNE. No novel methods or modeling algorithms are proposed, thereby limiting its relevance for the KDD audience.
3. What is the meaning of 'Ref' in Table 1?

---

### Decision · Program_Chairs · 2024-06-28

Accept (Poster)